# Deep Learning-Based Tumor Segmentation of Murine Magnetic Resonance Images of Prostate Cancer Patient-Derived Xenografts

**DOI:** 10.3390/tomography11030021

**Published:** 2025-02-22

**Authors:** Satvik Nayak, Henry Salkever, Ernesto Diaz, Avantika Sinha, Nikhil Deveshwar, Madeline Hess, Matthew Gibbons, Sule Sahin, Abhejit Rajagopal, Peder E. Z. Larson, Renuka Sriram

**Affiliations:** Department of Radiology and Biomedical Imaging, University of California-San Francisco, San Francisco, CA 94158, USA; satviknayak@gmail.com (S.N.); hsalkeverbaseball@gmail.com (H.S.); ernesto.diaz@ucsf.edu (E.D.); avantika.sinha@ucsf.edu (A.S.); nikhil.deveshwar@ucsf.edu (N.D.); madeline.hess@ucsf.edu (M.H.); matthew.gibbons@ucsf.edu (M.G.); abhe.rajagopal@alleninstitute.org (A.R.)

**Keywords:** prostate cancer, magnetic resonance imaging, patient-derived xenografts, deep learning, tumor segmentation

## Abstract

Background/Objective: Longitudinal in vivo studies of murine xenograft models are widely utilized in oncology to study cancer biology and develop therapies. Magnetic resonance imaging (MRI) of these tumors is an invaluable tool for monitoring tumor growth and characterizing the tumors as well. Methods: In this work, a pipeline for automating the segmentation of xenografts in mouse models was developed. T_2_-weighted (T2-wt) MRI images from mice implanted with six different prostate cancer patient-derived xenografts (PDX) in the kidneys, liver, and tibia were used. The segmentation pipeline included a slice classifier to identify the slices that had tumors and subsequent training and validation using several U-Net-based segmentation architectures. Multiple combinations of the algorithm and training images for different sites were evaluated for inference quality. Results and Conclusions: The slice classifier network achieved 90% accuracy in identifying slices containing tumors. Among the various segmentation architectures tested, the dense residual recurrent U-Net achieved the highest performance in kidney tumors. When evaluated across the kidneys, tibia, and liver, this architecture performed the best when trained on all data as compared to training on only data from a single site (and inferring on a multi-site tumor images), achieving a Dice score of 0.924 across the test set.

## 1. Introduction

Longitudinal in vivo studies of murine xenograft models are widely utilized in oncology to study cancer biology, specifically growth characteristics upon intervention and develop therapies. For subcutaneous tumor studies, measuring tumor growth is relatively straightforward through the use of a caliper. Since these tumors are implanted just beneath the animal’s skin, their volume can be easily measured externally with relatively good accuracy. However, subcutaneous tumors have the limitation of inconsistent growth caused by changes in metabolism, lack of uniform vascularization and delivery of nutrients, and development of necrosis as the tumor progresses. Tumors implanted in the kidney have much more consistent growth as the region is well-perfused, preventing tumors from becoming hypoxic or necrotic. These tumors can also be transplanted into other sites, such as the liver and bone, which are common sites of metastases and provide different micro-environments for studying cancer growth and treatment. Consequently, studies of tumors in the kidney, liver, and tibia are valuable in preclinical studies, but it is difficult to measure tumor volume, necessitating non-invasive imaging approaches. One often pursued solution to this is the use of magnetic resonance imaging (MRI). Tumor volume is measured based on the manual segmentation of the visible tumor regions over multiple slices from an MRI scan. By combining the tumor area in all the individual slices, the total volume can be calculated based on the resolution of MRI acquisition. Measuring the volume over time to determine growth rates can be used to test the efficacy of various interventions.

The tumor segmentation process is typically performed manually by researchers, making it extremely time intensive and requiring high levels of training in order to obtain accurate tumor volume measurements. It is also subject to potential inaccuracies from interuser variability as a result of researcher bias leading to a lack of reproducibility.

In attempts to solve this issue, both fully automated and semi-automated algorithms have been developed for tumor segmentation. Deep learning with convolutional neural networks (CNNs) has been highly successful for segmentation in nearly all major imaging modalities [1]. However, the vast majority of automated tumor segmentation using MRI images has been predominantly implemented for clinical datasets and almost exclusively for brain tumors [2]. There exist only a handful of preclinical MRI tumor segmentation applications, tailored to various tasks such as multicenter brain tumor segmentation [3], tibia segmentation [2], a myelofibrosis model [4], sarcoma [5], and triple-negative breast cancer xenografts [6].

The primary objective of our study is to develop an end-to-end pipeline using a deep-learning model to automatically measure volumes of xenograft tumors at various sites in mouse models using MRI data. An automated process for segmentation will save researchers a significant amount of time, ensuring reproducibility of results as well as increased accuracy in comparison to manual tumor segmentation. Our base starting point was work performed by Dr. Shoghi’s group at Washington University at St. Louis, who developed a dense recurrent residual U-Net model to delineate patient-derived breast cancer xenograft tumors placed in the mammary fat pad using both T1-weighted and T2-weighted (T2-wt) MRI images [6]. We adapted this work to apply to xenografts at multiple sites, including kidneys, liver, and tibia, to use only T2-wt MRI, and to be applied across multiple slices for volumetric segmentation.

## 2. Methods

### 2.1. Imaging Data Characteristics

Six different prostate cancer patient-derived xenografts (PDX) were utilized in this study. The castrate-resistant prostate cancer (CRPC) PDX included LTL313HR, LuCaP 70CR, and LuCaP 77CR and were experimentally derived castration-resistant sublines of androgen-dependent PDX [7,8]. The small cell neuroendocrine carcinoma (SCNC) PDX (LTL352, LTL610, LuCaP 93, and LuCaP 145.1) were derived from patients who underwent prolonged androgen-deprivation therapy [7,8]. The PDX were first implanted under the renal capsule of male NSG mice (either bred at our institution or procured from the Jackson Laboratory). Following sufficient tumor growth in the renal capsule, tumors were harvested, and upon single-cell digestion, were implanted into the liver or tibia of separate male NSG mice. All experiments on animals were conducted in compliance with the policies set forth by and with approval from the University of California, San Francisco (UCSF) Institutional Animal Care and Use Committee (IACUC).

The MRI images used in the study were acquired using a BioSpec 3T scanner (Bruker Billerica, MA, USA). The mice were placed supine, headfirst, into the scanner, and axial images were taken to cover the entire tumor region, including edge slices with no tumor. Two-dimensional T2-wt images were acquired using rapid acquisition with relaxation enhancement (RARE) sequence in PV3.6 using a 40 mm ^1^H quadrature transmit-receive volume coil. Interleaved slices of 1 mm thickness (without slice gap) were acquired to cover the entire tumor with TE = 48.0 ms, TR = 2763.0 ms, 2 averages, and an in-plane spatial resolution of 170–330 micron in-plane resolution. Mice were scanned serially to monitor tumor growth. No external contrast agent imaging was performed.

Manual segmentation of the tumors performed by experienced users was used to create tumor masks that represented the ground truth used in training and testing the model. The regions of interest (ROIs) were drawn primarily by 4 researchers with experience working with prostate cancer mouse models. All annotators were trained to draw the tumor ROIs by the same person (RS). They were instructed to identify tumor boundaries on T2-wt images through the following criteria: For kidney xenografts, the contralateral healthy kidneys had well-defined borders and were used as a reference. In contrast, tumors present with heterogeneous hyperintense signals relative to the contralateral kidney. For liver xenografts, healthy livers were isointense; tumors presented a slightly hyperintense signal. For tibia xenografts, the healthy tibia was used as a reference, and implanted tumors had a well-defined, distinct, heterogeneous, and hyperintense appearance. Areas of necrosis, which appeared hypointense relative to the tumor, were avoided. Annotations were performed using IDL-based software developed in-house (BRIMAGE, IDL 8.5.2, build Linux x86_64 m64) [9].

The total number of animals, scans, and 2D images are shown in Table 1. The data were split on an individual scan level as shown in Table 1. Note that with the five-fold cross-validation training method used, the training data set is split into training:validation at a 4:1 ratio (see more details below). The scans for the test set were selected to specifically include scans of each of the different PDX phenotypes and also to contain scans with both smaller and larger tumors. Representative volume ranges were included in the training dataset, with at least 2 datasets for each region in the tumor volume range of up to 0.1 cc, 0.1–0.5 cc, and 0.6 cc–1.2 cc. Images with obvious motion artifacts (streaking along the frequency encode direction) in kidney and liver tumors were excluded. A representative set of images of varying tumor volume in each of the inoculation sites is shown in Appendix A.

### 2.2. Image Preprocessing

Data used in this project were initially stored in the DICOM format for MRI. Manually segmented tumor masks were stored as .mir files and were created using the home-built IDL-based image processing tool BRIMAGE. All image data were converted from DICOM images to NumPy arrays, storing the original pixel values without the information from the DICOM header. Converting the manual tumor masks (.mir files) to NumPy arrays required first converting the .mir to .mat files via a MATLAB script and then converting the .mat files into NumPy arrays. All data used had image dimensions of 192 × 192 or were scaled up to 192 × 192. When scaling up the tumor masks, a thresholding value equal to the median in the scaled ROI was used to binarize the data. For the images, normalization to a range of 0 to 1 was performed on an image-to-image basis prior to data loading using the algorithm below (*p* represents a pixel in the image; *I* represents the entire image).$$p_{normalized}=\frac{p_{original}-\min\left(I\right)}{\max\left(I\right)-\min\left(I\right)}$$

The images were randomly augmented with translations horizontally and vertically up to a factor of 0.05 of the FOV, zoomed in and out by up to a factor of 0.05, and sheared by up to a factor of 0.05. We did not include flipping since our practice is to always place the xenografts in the left kidney and left leg, and this also encodes the asymmetry of the liver location. Image augmentations were applied to training data alone and not to the validation or test set.

### 2.3. Pipeline and Model Architectures

We propose a tumor volume measurement pipeline that includes multiple stages, as illustrated in Figure 1. The input to the pipeline is a single slice image, which is first fed through a slice classification model. Slices that are classified as including a tumor are fed through a segmentation model to predict a tumor mask. To process the entire tumor volume, the results of this pipeline are combined for all slices. Each component is described in the section below.

#### 2.3.1. Slice Classifier

A slice classifier was trained using a CNN, ResNet50, with an input of a T2-wt MRI slice (single image) and a binary output of “tumor” or “no tumor”. It was trained based on all of the training and validation images from the kidney, liver, and tibia datasets.

#### 2.3.2. Tumor Segmentation

We evaluated several models for tumor segmentation, all of which were in the U-Net family that consists of downsampling and upsampling to extract features from the input images as well as skip connections to produce a mask of the predicted tumor region. The model outputs are essentially a probability map with the same dimensions as the input images with a value between 0 and 1. If the value is greater than or equal to 0.5, the model predicts that the pixel corresponds to a tumor, and if the value is less than 0.5, the model predicts that the pixel does not correspond to a tumor. We compared a basic U-Net [10], attention U-Net [11], dense recurrent residual U-Net (D-R2UNet) [6], a transformer U-Net (UNetR) [12], and a state-space U-Net (UMamba) [13].

We began by experimenting with a basic U-Net architecture. Due to the relatively small size of the available images in our dataset, we hypothesized that a model with fewer parameters may generalize better than the deeper models. We also sought a baseline of performance that we could compare the accuracy of the other variants against. The skip connections in U-Net style architectures are beneficial in alleviating vanishing gradient issues; however, incorporating them into the decoder can introduce information that may not be relevant to high-level semantic understanding. We chose to implement an attention U-Net in the interest of developing a relatively parameter-efficient model where we can selectively incorporate meaningful skip connections information through soft attention. We tested the DR2-UNet architecture for its high adaptability to various tumor phenotypes. It combines attributes found in basic U-Net architectures with dense and residual connections. Although it involved significantly more parameters than others we tested, we also implemented a UNetR architecture, a U-Net that replaces the encoder–downward convolution with a vision transformer model. The transformer encoder introduced the ability to extract more multi-scale information from the images, as opposed to the basic and attention U-Net, which were confined to their local frames. We then chose to implement a UMamba model as a more parameter-efficient alternative to the UNetR. We specifically chose the LightM-UNet variant, as it had been shown to achieve similar performance to the UNetR at a parameter count of only roughly 1 million [14].

### 2.4. Training

We compared several loss functions, including the Dice score, binary cross-entropy, and boundary loss. The Dice score is generally considered to be the gold standard metric for evaluating segmentation accuracy. However, the use of this metric cannot effectively reward model predictions where the ground truth has no tumor. We obtained the best results using a combination of the dice loss and binary cross-entropy loss. We chose to use binary cross-entropy because it is effective in assessing classification performance [15]. Our dataset was heavily skewed towards false values since the tumor regions took up significantly less space compared to the nontumor regions in our images. To account for this, we modified the binary cross-entropy function to weight true and false values equally. This helps prevent the background pixels from skewing our loss value due to their prevalence compared to tumor pixels. A set weight was assigned to each loss, and the losses were summed after being multiplied by the weight, expressed as $Composite\;Loss=\omega_{Dice}\times Dice+\omega_{BCE}\times BCE$. The process of determining the optimal weights was based on a series of experiments trying different ratios. Based on prior work of others in the field, the Dice score is considered a well-regarded metric for segmentation. Since our main goal was creating an accurate segmentation of a region, rather than the classification based on tumor prevalence in an image, we decided it would be most beneficial to include at least 0.5 weight for Dice loss. We performed trials using Dice/BCE ratios of 50/50, 75/25, 90/10, and 100/0. From our initial experimentation, we ultimately, chose weights of $\omega_{Dice}=0.75,\;\omega_{BCE}=0.25$.

Prior to the final model training, we performed a series of experiments with different combinations of training parameters. We tested three optimizers, four learning rate schedulers, six learning rates, and seven batch sizes. For optimizers, we used the Adam [16], AdamW [17], and SGD (gradient descent) optimizers [18]. For SGD, we tested different momentum values: 0 (no momentum), 0.25, 0.5, 0.75, 0.9, and 0.99. We experimented with six base learning rates that were all powers of 10: 0.1, 0.01, 0.001, $1\times 10^{-4}$, $1\times 10^{-5}$, and $1\times 10^{-6}$. Instead of using a constant learning rate throughout the entire model training, we tested various schedulers that adjust the learning rate throughout model training. We utilized 3 schedulers from the TensorFlow Keras library: cosine decay, cosine decay with restarts, and exponential decay. We also custom wrote a linear learning rate scheduler that continuously decreases the model’s learning rate from the initial learning rate linearly down to 0 (available on GitHub repository). Additionally, we tried using different batch sizes. We selected the 10 best out of 15 models from a combination of the optimizer, learning rate, and scheduler and tested seven batch sizes for each model using powers of 2 ranging from 1 to 64. The results of these experiments can be found in the Appendix A. Briefly, we achieved optimal results using a stochastic gradient descent optimizer at 50% momentum, beginning with an initial learning rate of 0.01 and gradually decreasing the learning rate linearly.

For final training, the models were trained for 250 epochs with a batch size of 2 and a fixed learning rate of $1\times 10^{-5}$. It took approximately 30 h to train the D-R2UNet model and 6 s to run inference on the test data on a Nvidia RTX A6000 graphics processing unit (GPU, Santa Clara, CA, USA). During training, each of the model’s approximately 27,000,000 nodes were continuously adjusted through backpropagation to minimize the value of the composite loss function on the training data. After each epoch in model training, the algorithm ran validation where it generated predictions based on the validation set and calculated the accuracy relative to the ground truth. The model was instructed to save the new set of weights only if there was an improvement in the validation composite loss. This was performed as one measure to prevent the model from overfitting the training data.

The D-R2UNet model was trained using K-fold cross-validation, as in prior work [6]. Five-fold cross-validation was utilized to generate five separate models. The training dataset was randomly split into five groups of images, and in each of the five iterations, one group was selected as validation data while the other four groups were used for training. K-fold cross-validation is beneficial by reducing the variance that may occur from a single train-validation split of the data. From the five models, the best performing model was selected by the model with the highest test Dice score.

## 3. Results

### 3.1. Tumor Slice Classifier

The first step in our pipeline is a slice classifier to identify the slices that contain part of the tumor, and only those slices are segmented. We achieved 89.6% accuracy with the CNN (with 2,378,369 parameters) and 89.9% accuracy with ResNet50 (which had 23,583,489 parameters) in identifying slices that contained tumors. These models used images from all xenograft locations (kidney, liver, tibia). The confusion matrix is shown in Table 2. The CNN had the advantage in that it required about an order of magnitude fewer parameters than the ResNet50 model.

### 3.2. Segmentation Architecture Comparison

The various segmentation model architectures tested are compared in Table 3 and in Figure 2. Overall, the D-R2UNet and U-Net models were the best performing models with near equivalent performance as measured by the Dice score. The attention U-Net and UMamba performance was not far behind, whereas the UNetR showed very poor performance on our dataset.

The example segmentations support the Dice score results, where the U-Net, attention U-Net, D-R2UNet, and UMamba all show high-quality segmentations that capture many of the shape features of the kidney xenograft tumors. The UNetR segmentation failures are visible in these examples (Figure 2).

### 3.3. Extension to Multiple Anatomical Sites

We evaluated the generalization of the segmentation across xenograft anatomical sites (Figure 3, Table 4). Figure 3 demonstrates representative images of each anatomical site and the performance of the dense recurrent residual U-Net. The near-perfect match of the trained model relative to the ground truth is evidenced by the lack of any marking in the difference image except in traces in some edges. For this, the D-R2UNet was used as it was the highest performing segmentation model when tested on the kidney xenograft data. First, we performed inference of the model (trained on kidney images alone) on the liver and tibia hold test sets. We noticed promising results with a dice score above 0.8 for both anatomies. We theorized that we could improve performance by creating models trained on each individual anatomy. Consequently, we trained three additional models: one trained exclusively on liver data, one trained exclusively on tibia data, and another trained on data from all three anatomies. As expected, the models performed best on the data they were trained on. The tibia xenograft segmentation performed relatively well regardless of training data, while the liver was the most challenging. The best overall performance in all cases came from a model trained with data from all anatomical sites.

## 4. Discussion

In this study, we developed a pipeline to automate the tumor segmentation process for prostate cancer PDXs in mice kidney, tibia, and liver. Our approach aimed to perform volumetric segmentation using T2-wt MRI as input. The key features of our approach were to first use a slice classifier to identify slices containing tumors for feeding into an image segmentation network. And second, a combined model capable of segmenting tumors reliably in different anatomical sites.

This project was inspired by work performed previously that implemented the dense recurrent residual U-Net model architecture for segmenting breast cancer [4]. In our work, we implemented some key changes to better suit the model to the needs of our lab’s preclinical work. The most significant difference was that we shifted from only a single model to utilizing a two-step process consisting of a classification model and a segmentation model. This is critical for optimal workflow to avoid having to manually exclude the slices adjoining the tumor, which is always acquired to ensure full tumor coverage. Our aim was for the pipeline to be able to receive a complete MR scan as input and automatically determine the total volume of the tumor.

The decision to implement a two-step model including a classifier was motivated by high-performance metrics on individual images containing tumors and with very low-performance metrics on images without tumors. This resulted in lower average performance of the pipeline and thus necessitated the addition of a classification model prior to segmentation. Despite this addition, we continued to include images without tumors in our training data set, enabling the segmentation model to predict the absence of a tumor even if it was not correctly predicted by the classifier. This decision effectively created two places in the process where images not containing tumors can be identified. We felt it was important to train the segmentation model on images both including and not including tumors since a critical role is to accurately delineate smaller tumors. Training on images not containing tumors shows the model that the tumor could be of any size, preventing it from forcing a segmentation when it does not exist.

Another distinct feature of our approach was the use of a single contrast image, namely T2-weighting. No T1-weighting or injected contrast agents were employed. We believe the high prevalence of features found in T2-weighted MR images warrants no need for additional information found in T1-weighted MR images. It was also shown in a murine sarcoma model that CNN-based segmentation results indicated that Dice scores were similar when using multicontrast data versus single T2-weighted data [5]. Additionally, using only a single image sequence also simplifies the preclinical workflow and increases efficiency, as often these are part of the minimalistic screening process ( for reduced imaging time and thereby cost) to monitor the tumor growth until an optimal size for metabolic imaging and intervention experiments. This also affords reduced inference times.

Furthermore, we were able to generalize to multiple prostate cancer PDXs and at multiple anatomical sites, so we utilized MR scans from 6 different PDXs in the kidney, liver, and tibia.

Within the model training itself, we did not apply any image augmentations to the validation data because we wanted the validation metrics to function as a predictor of how the model will perform during inference. Additionally, the training data were only augmented 75% of the time in order for the model to better converge onto the training data by viewing the original images more often during training. This implementation allowed for the unmodified images to be shown more often to the model during training, rather than only augmented images.

During model training, we choose to utilize five-fold cross-validation training in order to experiment with a variety of training/validation data splits. This ensures that all training images have the opportunity to be part of the training and validation process. The performance metrics reported reflect the models that performed the best on the holdout test set.

We noticed some interesting patterns when comparing our inference results for the models trained on data for the individual anatomical sites. All models seemed to generalize well to the tibia dataset but had difficulty making accurate predictions for the liver dataset. The combined model, trained on data from all three anatomical sites, had the best performance across the board. This indicates an ability for the model to identify shared patterns across multiple sites. Additionally, although the same PDX across multiple inoculation sites is used, it is important to note that an intact tissue slice is implantedin the kidney while single cell digested material is injected in the liver and bone. This is an indication of the model’s capacity to learn the common characteristics of a tumor, but it requires some individualized data to understand what a tumor in a given site should look like. We predict that utilizing additional data from different anatomical sites should result in more generalizability of the model and its application.

Some of the shortcomings of this work include training only on prostate cancer data, which includes six different PDXs of two different phenotypes. While we have demonstrated the performance of our pipeline on the different sites of tumor growth, its performance on other cancers and other regions of growth remains to be tested. Owing to the use of prostate cancer PDX, our data set is limited to male immunocompromised mice of 4–8 months of age. The inclusion of female mice would be important to evaluate in the context of generalized models in addition to the cancer types. Additionally, we made the decision to prioritize a larger training set rather than a more common train-test split. This decision was made in an attempt to provide the model with as much information as possible during training. In order to compensate for this, we selectively chose a representative set of T2-wt scans from each PDX, including a distribution of tumor sizes to be included in the test set.

## 5. Conclusions

In this work, we developed a pipeline for automatic segmentation of tumor xenografts in multiple anatomic sites using seven different PDX models of two different phenotypes using a single model. We are currently in the process of deploying this for preclinical cancer researchers for easier and faster tumor volume estimation using MRI using a graphical interface. In the future we hope to incorporate MRI images of other cancer types and anatomical locations for a universal model for automatic tumor segmentation. These approaches would provide an indispensable tool for preclinical researchers to evaluate drug response in a more robust and optimal methodology.

## Figures and Tables

**Figure 1 tomography-11-00021-f001:**
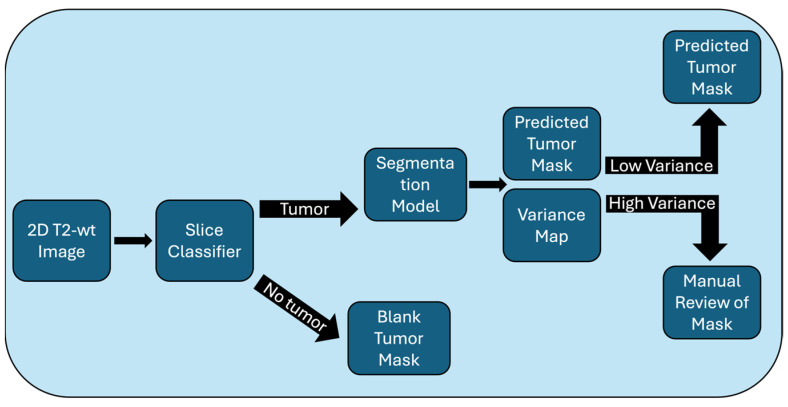
(**top**) Illustration of the xenograft MRI segmentation pipeline proposed. This includes a slice classification model, a tumor segmentation model, and a criteria for identifying uncertainty in the segmentation model. (**bottom**) Schematic of the dense recurrent residual U-Net (D-R2UNet) architecture that was the top-performing model in this and previous xenograft segmentation work [6].

**Figure 2 tomography-11-00021-f002:**
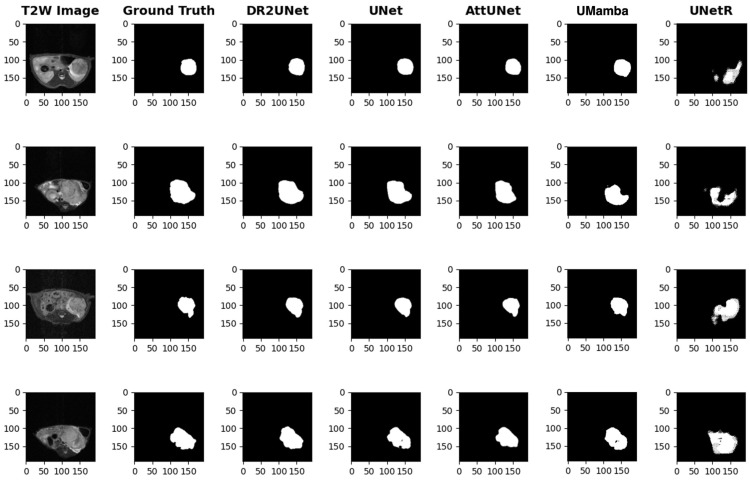
Sample segmentation results for the various image segmentation model architectures. T2-wt MRI slices from four different mice with kidney xenografts are shown, and these depict a range of tumor shapes and surrounding anatomy/contrast that is typical of our kidney xenograft dataset.

**Figure 3 tomography-11-00021-f003:**
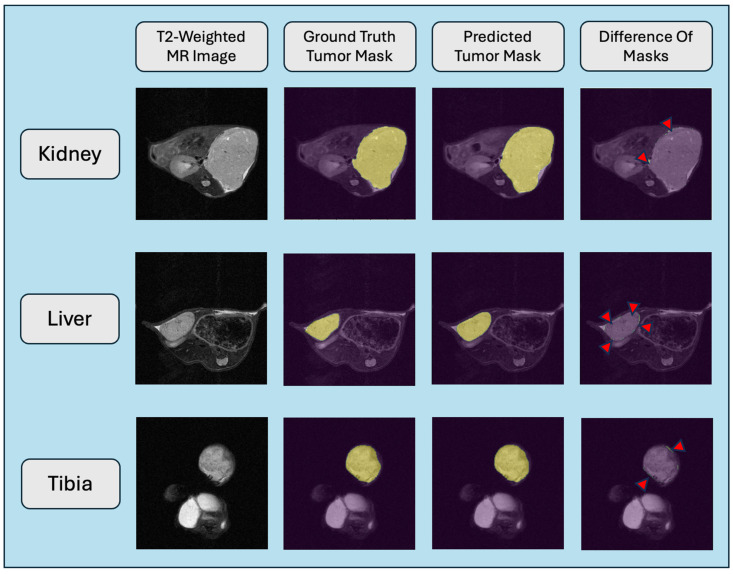
Representative single slice results from inference of the trained dense recurrent residual U-Net model across xenograft anatomical sites. Models trained only on data from the individual anatomical site datasets are shown here. The red arrows in the difference masks indicates the trace mismatch that occurs at the edges of tumor from the automated segmentation.

**Table 1 tomography-11-00021-t001:** Number of datasets used in this project. The anatomical region indicates where the xenograft was located in the mice.

Anatomical Region	Total Number of Mice	Number of Scans	Total 2D Images	2D Images Containing a Tumor	Training/Test (Scans)	Training/Test (Images)
Kidney	34	128	2558	1513	121/7	2416/142
Liver	39	142	3135	1749	134/8	2955/180
Tibia	25	106	2337	1332	100/6	2205/132

**Table 2 tomography-11-00021-t002:** Performance for tumor slice classifiers on the test set is shown as a confusion matrix.

Convolutional Neural Network	Predicted Tumor	Predicted No Tumor
**Actual Tumor**	303	29
**Actual No Tumor**	18	104
**Resnet50**	**Predicted Tumor**	**Predicted No Tumor**
**Actual Tumor**	303	29
**Actual No Tumor**	17	105

**Table 3 tomography-11-00021-t003:** Comparison of performance for the segmentation models with different architecture on the kidney xenograft dataset.

	U-Net	Attention U-Net	UNetR	UMamba	D-R2UNet
Dice score	0.912	0.894	0.633	0.866	0.914

**Table 4 tomography-11-00021-t004:** Comparison of Dice scores across all anatomical site datasets (kidney, liver, tibia) for segmentation models trained on various combinations of the data. Note: these were computed only on slices that were correctly classified as containing a tumor by the slice classifier.

		Training Dataset			
		D-R2UNet Kidney	D-R2UNet Liver	D-R2UNet Tibia	D-R2UNet Combined
Inference Dataset	Kidney Dice	0.914	0.855	0.806	0.921
	Liver Dice	0.785	0.897	0.666	0.911
	Tibia Dice	0.916	0.846	0.919	0.945
	Combined Dice	0.873	0.872	0.816	0.924

## Data Availability

Dataset available on request from the authors.

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
