# Peer review of "Deep Learning-Based Tumor Segmentation of Murine Magnetic Resonance Images of Prostate Cancer Patient-Derived Xenografts"

_tomography, 2025, doi:10.3390/tomography11030021_

Round 1

Reviewer 1 Report

Comments and Suggestions for Authors

This paper presents a method based on deep learning aimed at segmenting tumors of prostate cancer in mice, but there are still some aspects that need improvement.

1.The novelty of the proposed method is not well-established. The use of a slice classifier followed by a segmentation network is common in the field.

2. For deep learning architecture, the evaluation of different U-Net architecture seems to be quite tokenistic, and it is difficult to draw meaningful conclusions about the superiority of the chosen D-R2UNet architecture.

3. The author combined loss functions in a way that seems a bit random. They should explain how they decided on those specific weights.

4. There is no information on some of the most critical parameters in MRI like the pulse sequences used, voxel size, number of averages, contrast agents, etc.

5. The authors barely mentioned how they processed the images. They should talk about how they addressed noise and movement artifacts, frequent issues seen in mouse MRI.

6. The manuscript didn't talk about any potential problems or limitations of their method.

7. The flowchart is difficult to read due to the small size of the font.

8. There was no information on the total number of animals used in the experiment and how many animals correlate with each prostate cancer patient-derived xenograft.

Author Response

Comment 1: The novelty of the proposed method is not well-established. The use of a slice classifier followed by a segmentation network is common in the field. 

Response 1: I wasn’t able to find any references after searching for them. This is likely because the search associates the keyword classifier with identifying the phenotype of a tumor rather than its presence. If we can find a direct reference, then it should be include it as part of how we chose the methodology. 

Comment 2: For deep learning architecture, the evaluation of different U-Net architecture seems to be quite tokenistic, and it is difficult to draw meaningful conclusions about the superiority of the chosen D-R2UNet architecture. 

Response 2: Our rationale for the choice of different architectures chosen has now been elaborated in section 2.3.2 

Comment 3: The author combined loss functions in a way that seems a bit random. They should explain how they decided on those specific weights. 

Response 3: Agree that more should be included about how this distribution was decided. It was not the most methodological process, so we added the following text to the discussion section describing how we arrived at this ratio. 

“The process of determining the optimal weightages was based on a series of experiments trying different ratios. Based on prior work of others in the field, the Dice score is a considered a well-regarded metric for segmentation. Since our main goal was creating an accurate segmentation of a region, rather than the classification of if an image contains a tumor, we decided it would be most beneficial to include at least 0.5 weight for Dice loss. We performed trials using Dice/BCE ratios of 50/50, 75/25, 90/10, and 100/0. From our initial experimentation, we ultimately, we chose weights of ?????=0.75,????=0.25?Dice=0.75,?BCE=0.25.” 

Comment 4: There is no information on some of the most critical parameters in MRI like the pulse sequences used, voxel size, number of averages, contrast agents, etc. 

Response 4: We agree that the inclusion of this information would be beneficial to the users. As mentioned only T2 weighted images were used for tumor segmentation. Further details have been added as shown below. 

“Interleaved slices of 1 mm thickness (without slice gap) were acquired to cover the entire tumor with TE = 48.0 ms, TR = 2763.0 ms, 2 averages and in-plane spatial resolution of 170-330 micron in-plane resolution. Mice were scanned serially to monitor tumor growth. No external contrast agent imaging was performed.” 

Comment 5: The authors barely mentioned how they processed the images. They should talk about how they addressed noise and movement artifacts, frequent issues seen in mouse MRI. 

Response 5: Agree that we didn’t include extensive information of the processing of the images during acquisition of the MRI. Standard processing of Bruker PV6.3 was used. Images with obvious motion artifacts (streaking along the frequence encode direction) in kidney and liver tumors were excluded. 

In regards of preprocessing of the images prior to inputting them into the model, we have added the following: 
“For the images, normalization to a range of 0 to 1 was performed on an image-to-image basis prior to training/data loading using the algorithm below (p represents a pixel in the image; I represents the entire image).” 

???????????= (?????????−min(?)) / (???(?)−???(?))

Comment 6: The manuscript didn't talk about any potential problems or limitations of their method. 

Response 6: We agree since this was a critical oversight on our end. We have modified our discussion by adding the following: 
“Some of the shortcomings of this work include training only on prostate cancer data which includes six different PDXs of two different phenotypes. While we have demonstrated the performance of our pipeline on the different sites of tumor growth, its performance of other cancers and other regions of growth remains to be tested. Owing to the use of prostate cancer PDX, our data set is limited to male immunocompromised mice of 4-8 months of age. The inclusion of female mice would be important to evaluate in the context of generalized models in addition to the cancer types. Additionally, we made the decision to prioritize a larger training set rather than a more common train-test split. This decision was made in an attempt to provide the model with as much information as possible during training. In order to compensate for this, we selectively chose a representative set of T2-wt scans from each PDX, including a distribution of tumor sizes to be included in the test set.” 

Comment 7: The flowchart is difficult to read due to the small size of the font. 

Response 7: Agree, we modified the flow chart and made the model architecture figure larger. All font sizes are now at least 12 point. 

Comment 8: There was no information on the total number of animals used in the experiment and how many animals correlate with each prostate cancer patient-derived xenograft. 

Response 8: Agree, this information will be valuable, so we added a column with this information to Table 1. 

Reviewer 2 Report

Comments and Suggestions for Authors

The paper shows a deep learning-based approach for automating the segmentation of prostate cancer patient-derived xenografts in mouse models using T2-weighted MRI data. The study develops a pipeline that includes a slice classifier network and tests various segmentation architectures, with a dense residual recurrent U-net achieving the highest performance across different implantation sites.

This is a well-structured and innovative study that addresses an important challenge in preclinical cancer research.  There are, however, some observations to be done before the publication of this manuscript.

Title:

Spell out all acronyms the first time that they are used (e.g.: MRI).

Abstract:

Again, spell out all acronyms the first time that they are used (e.g.: MRI).

Please, add more specific numerical data.

Methods:

The authors should provide more details on the MRI acquisition parameters, explain the rationale for choosing the specific architectures tested and include information on data augmentation techniques, if any were used.

Discussion:

Explore potential applications of this technology in drug development or personalized medicine.

Author Response

Comment 1: Title: Spell out all acronyms the first time that they are used (e.g.: MRI). 

Response 1: The title has been updated to avoid acronyms. 

Comment 2: Abstract: Again, spell out all acronyms the first time that they are used (e.g.: MRI). Please, add more specific numerical data. 

Response 2: Thank you. The acronyms have been spelled out, including the title. 

Comment 3: Methods: The authors should provide more details on the MRI acquisition parameters, explain the rationale for choosing the specific architectures tested and include information on data augmentation techniques, if any were used. 

Response 3: We agreed that this information would be beneficial to include. The MRI acquisition parameters have been updated. Information of the data augmentation used is located in the second paragraph of section 2.2. We added the following to explain the rational for the architectures chosen: 
“We began by experimenting with a basic U-Net architecture. Due to the relatively small size of the available images in our dataset, we hypothesized that a model with fewer parameters may generalize better than the deeper models. We also sought a baseline of performance that we could compare the accuracy of the other variants against. The skip connections in U-Net style architectures are beneficial in alleviating vanishing gradient issues; however, incorporating them into the decoder can introduce information that may not be relevant to high-level semantic understanding. We chose to implement an Attention U-Net in the interest of developing a relatively parameter-efficient model where we can selectively incorporate meaningful skip connections information through soft attention. We tested the DR2-UNet architecture for its high adaptability to various tumor phenotypes. It combines attributes found in basic U-Net architectures with dense and residual connections. Although it involved significantly more parameters than others we tested, we also implemented a UNetR architecture, a U-Net that replaces the encoder-downward convolution with a vision transformer model. The transformer encoder introduced the ability to extract more multi-scale information from the images, as opposed to the basic and attention U-Net, which were confined to their local frames. We then chose to implement a UMamba model as a more parameter-efficient alternative to the UNetR. We specifically chose the LightM-UNet variant, as it had been shown to achieve similar performance to the UNetR at a parameter count of only roughly 1 million [14].” 

Comment 4: Discussion: Explore potential applications of this technology in drug development or personalized medicine. 

Response 4: This work is exclusively focused on preclinical tumor models which are an indispensable tool for drug development. 

Reviewer 3 Report

Comments and Suggestions for Authors

This paper presents a deep learning-based tumor segmentation pipeline designed for MR images of PDX specific for prostate cancer in mouse models. The proposed pipeline automates tumor volume measurements in various anatomical regions, including the kidney, liver, and tibia. The segmentation process includes a slice classification model that processes T2-weighted MR images followed by a tumor segmentation model. D-R2UNet achieved the highest performance among the tested architectures with a reported Dice score of 0.924. This work aims to improve the reproducibility of tumor measurements while significantly reducing the time and effort required for manual segmentation.

By proposing a deep learning-based approach, this study makes a valuable contribution to preclinical research by addressing challenges such as segmentation accuracy and consistency. The model's generalizability is demonstrated by applying it to tumors in different anatomical regions. Moreover, the detailed description of data preprocessing, model architectures, and optimization strategies enhances the reproducibility of the methodology. This work can improve clinical workflows by monitoring tumor growth and evaluating treatment efficacy by providing an efficient and reliable segmentation tool.

However, I kindly suggest that the authors take into account my following gentle suggestions:

1- Using data from wage models with different age, gender, and genetic variations may contribute to a broader generalizability of the model.

2- The study could better emphasize the model's effectiveness by more comprehensively comparing its results with manual segmentation methods.

3- A discussion on the applicability of the study to clinical settings and potential challenges should be added.

4- A detailed analysis of the segmentation errors and how to minimize them should be discussed.

5- More references in this field should be added.

6- Figure 3 and Table 4 should be explained in detail.

Author Response

Comment 1: Using data from wage models with different age, gender, and genetic variations may contribute to a broader generalizability of the model. 

Response 1: We agree. This would be invaluable to evaluate. However, given the limitations of the data available, such as the use of patient-derived xenografts of prostate cancer model, the current work is limited to male immunocompromised mice 3-6 months of age (owing to the wide range of tumor growth characteristics). We do not have readily available access to additional data since for this project we did not directly generate any new data, instead we utilized existing data that was generated for previous studies. In future work this will be expanded to include different mice genders, ages, and genetic variations.

Comment 2: The study could better emphasize the model's effectiveness by more comprehensively comparing its results with manual segmentation methods. 

Response 2: We agree that we need a more thorough comparison of our results with manual segmentation methods. We will perform this by having a series of MRI get annotated by multiple researchers and write an algorithm to compare the accuracy of manually generated segmentations with the results from the model. 

Comment 3: A discussion on the applicability of the study to clinical settings and potential challenges should be added.

Response 3: Agreed, we have discussed the shortcomings of this work in Discussion.

Comment 4: A detailed analysis of the segmentation errors and how to minimize them should be discussed. 

Response 4: We currently do not have the resources available in order to do this since for each MRI scan, we only have 1 or 2 sets of annotated tumor regions. To include this change, we would need to have more researchers go back and create additional annotations so we can analyze the difference between the manually created annotations and the model’s predictions. 

Comment 5: More references in this field should be added. 

Response 5: Agreed, we have added more references.

Comment 6: Figure 3 and Table 4 should be explained in detail. 

Response 6: Descriptive details have been included.

Round 2

Reviewer 1 Report

Comments and Suggestions for Authors

The authors have addressed the concerns adequately. I recommend the manuscript for publication.

Reviewer 2 Report

Comments and Suggestions for Authors

The revised manuscript along with the responses have addressed my concerns well. I have no more comments.